# ADVERSARIAL TRAINING WITH PERTURBATION GENERATOR NETWORKS

## ABSTRACT

Despite the remarkable development of recent deep learning techniques, neural networks are still vulnerable to adversarial attacks, i.e., methods that fool the neural networks with perturbations that are too small for human eyes to perceive. Many adversarial training methods were introduced as to solve this problem, using adversarial examples as a training data. However, these adversarial attack methods used in these techniques are fixed, making the model stronger only to attacks used in training, which is widely known as an overfitting problem. In this paper, we suggest a novel adversarial training approach. In addition to the classifier, our method adds another neural network that generates the most effective adversarial perturbation by finding the weakness of the classifier. This perturbation generator network is trained to produce perturbations that maximize the loss function of the classifier, and these adversarial examples train the classifier with a true label. In short, the two networks compete with each other, performing a minimax game. In this scenario, attack patterns created by the generator network are adaptively altered to the classifier, mitigating the overfitting problem mentioned above. We theoretically proved that our minimax optimization problem is equivalent to minimizing the adversarial loss after all. Beyond this, we proposed an evaluation method that could accurately compare a wide-range of adversarial training algorithms. Experiments with various datasets show that our method outperforms conventional adversarial training algorithms.

## 1 INTRODUCTION

Deep learning has shown the impressive performance in all areas of artificial intelligence, such as image classification and speech recognition (Hinton et al., 2012; Krizhevsky et al., 2012). These advances lead to a broad application of deep neural networks in various real-life tasks. There are still, however, severe security issues such as adversarial examples, which hinder the use of machine learning system until a complete defense is constructed against multiple adversarial attacks. Adversarial examples are data samples that are close to real data samples, which cause a given neural network to misclassify. The basic idea of adversarial examples is to find a sample that increases the loss value of a neural network in the neighborhood of training data (Szegedy et al., 2014). The perturbation on the original training data is so small that it makes the adversarial examples indistinguishable from the original examples.

Many authors proposed methods that make neural networks robust to adversarial examples (Papernot et al., 2016; Goodfellow et al., 2015; Szegedy et al., 2014; Miyato et al., 2016). One of the methods is an adversarial training, which re-trains the neural network with adversarial examples generated by adversarial attacks. Adversarial training with powerful attacks would guarantee robustness, but the recent fatal attack methods (Szegedy et al., 2014; Papernot et al., 2016; Carlini & Wagner, 2017; Moosavi-Dezfooli et al., 2016) require high computational complexity because of their iterative optimization. Therefore, they are not compatible with adversarial training. Methods that quickly produce adversarial examples, such as fast gradient sign (Goodfellow et al., 2015) or projected gradient descent (Kurakin et al. (2017); Madry et al. (2018)), have been used for practical adversarial training. While the above adversarial training methods are empirically successful, they might be susceptible to future attackers, and this makes the defense procedure useless. If an algorithm for generating an adversarial example is fixed in adversarial training, the network could overfit to the specific algorithm.

In this paper, we introduce a novel adversarial training framework that increases the robustness against various adversarial attacks. Stemming from GAN framework, we devised a method in which the classifier network and a *perturbation generator network* are alternately trained. To be more specific, the generator network generates a perturbation image that maximizes the loss function of the classifier network, and the classifier network is trained through the corresponding adversarial image with the true label. Through this minimax optimization between the two networks, the classifier network can improve robustness against many different attacks, as the attack pattern of the generator network is constantly modified depending on the classifier network. This procedure can be used in practical adversarial training since adversarial perturbations can be produced by a forward-propagation. We generalized Madry et al. (2018)'s research on adversarial loss to theoretically support our technique, and we also proposed a method that can fairly evaluate the performance of adversarial training algorithms.

## 2 RELATED WORKS

The goal of our work is to construct defensive mechanisms to adversarial attacks. To alleviate the security problem, the adversarial robustness of neural networks has been studied in the literature. One of the intuitive ways to increase robustness is to re-train with adversarial examples, which are called adversarial training. This method uniformly smoothen the ground-truth label decision region close to the original data points. In the context of smoothness, there exists adversarial examples that hold very low confidence on the ground-truth label in the vanilla decision region before applying robust optimization.

Szegedy et al. (2014) first proposed a method to generate adversarial examples. They use box-constraint L-BFGS optimization to find the examples. This holds the exact formulation of adversarial examples, but because of its exhausted optimization procedure, it is not suitable for practical adversarial training. Goodfellow et al. (2015) introduced an algorithm that quickly generates adversarial examples by using one-step gradient update, which is called fast gradient sign method. In addition, they first proposed a realistic adversarial training method which injects the adversarial examples into the training data. This method is not strong enough to generate high-quality examples and is far from robust optimization. Kurakin et al. (2017) suggested an iterative version of fast gradient method (FGM) attack called Projected Gradient Descent (PGD), which is much closer to the optimal adversarial examples. Adversarial training can be formulated with the robust optimization problem which minimizes the loss of the optimal adversarial examples in the $\epsilon$-ball of all the original data points. This gives rise to the following minimax game, which is the main theoretical background of our work:

$$\min_{\theta} \rho(\theta), \quad \text{where} \quad \rho(\theta) = \mathbb{E}_{(x,y) \sim \mathcal{D}} \left[ \max_{\delta \in S} L(\theta, x + \delta, y) \right]. \tag{1}$$

They approximated the above minimax game by PGD based adversarial training to reduce computational complexity issue. The gradient descent based adversarial examples for robust optimization is not adaptive. Therefore, those neural networks are vulnerable to other types of adversarial attacks (Athalye et al., 2018).

Several works studied the methods that generate stronger adversarial attacks(Athalye et al., 2018; Lee et al., 2017; Papernot et al., 2017; Moosavi-Dezfooli et al., 2016; Dong et al., 2018; Song et al., 2018). Carlini & Wagner (2017) pin-points that defensive distillation network (Papernot et al. (2016)) is not practical in that it exploits gradient masking, so they devised a powerful attack algorithm that avoids this problem. In an attempt to eliminate the gradient masking problem of softmax function, they adopted logits $Z$ in objective function, and discovered an appropriate adversarial noise for each image utilizing line-search technique. However, most of these works have high computational cost, so they are difficult to be applied to adversarial training.

There are many other defense methods that are not based on adversarial training (Li et al., 2019; , Junbo). The above robust optimization problem can be generalized as convex outer adversarial polytope, which relaxes the activation function as a convex form to prevent misclassification (Wong & Kolter, 2018). Certified defense algorithms guarantee at least a certain bounds of the proper label probability distributions against adversarial examples (Cohen et al., 2019; Liu et al., 2019; Raghu-

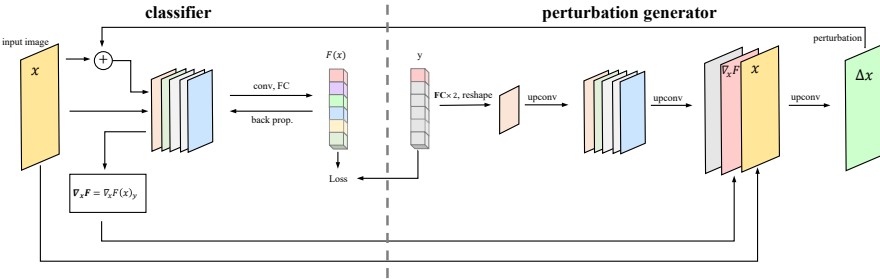

Figure 1: Entire procedure of our method: Conventional convolutional neural network is used as the classifier. The perturbation generator network receives a one-hot encoded label as input, which is processed with fully connected and up-convolutional layer, concatenates with the gradient image and the original image, and finally generates an adversarial perturbation.

nathan et al., 2018). In addition, some researchers recently studied the theoretical backgrounds of adversarial robustness (Dohmatob, 2019; Wang et al., 2018; Roth et al., 2019).

## 3 PROPOSED METHOD

### 3.1 NOTATIONS

We denote a labeled training set by $(\boldsymbol{x}, \mathrm{y}) \sim P_{\mathrm{data}}$, where $\boldsymbol{x} \in \mathbb{R}^{H \times W \times C}$ represents input images with height $H$, width $W$, and channel $C$, and $\mathrm{y} \in \{1, 2, \ldots, K\}$ is a label for an input $\boldsymbol{x}$. We use two neural networks in the proposed method. One is a standard $K$-class classifier network $F(\boldsymbol{x}; \boldsymbol{\theta})$ which is defined by:

$$F : \mathbb{R}^{H \times W \times C} \to \mathbb{R}^K, \quad F(\boldsymbol{x}; \boldsymbol{\theta}) = [F(\boldsymbol{x}; \boldsymbol{\theta})_1, F_\theta(\boldsymbol{x}; \boldsymbol{\theta})_2, \ldots, F(\boldsymbol{x}; \boldsymbol{\theta})_K]^T \quad (2)$$

Where $F(\boldsymbol{x}; \boldsymbol{\theta})$ represents the class probability vector computed using the softmax function. The other is a perturbation generating network $G(\boldsymbol{\nabla_x} F, \boldsymbol{x}, \mathrm{y}; \boldsymbol{\phi})$, which is defined by:

$$G : \left( \mathbb{R}^{H \times W \times C}, \mathbb{R}^{H \times W \times C}, \mathbb{R} \right) \to \mathbb{R}^{H \times W \times C} \quad (3)$$

Note that $G(\boldsymbol{\nabla_x} F, \boldsymbol{x}, \mathrm{y}; \boldsymbol{\phi})$ represents the perturbation of the input image $\boldsymbol{x}$, where $\boldsymbol{\nabla_x} F = \nabla_x F(\boldsymbol{x}; \boldsymbol{\theta})_\mathrm{y}$ denotes the gradient of class probability of the true label with respect to the input images $\boldsymbol{x}$.

### 3.2 ADVERSARIAL TRAINING WITH GENERATIVE MODEL

The entire procedure of our algorithm is shown in Figure 1. Goodfellow's work on GAN inspired us to make the classifier and the perturbation generator compete with each other. Classifier $F$ defines the network we are aiming to train and increase the robustness, and the perturbation generator $G$ is the network which produces the perturbations that maximize the loss function of the classifier. The classifier network is trained with adversarial images produced by the generator network with the true label. The generator network assigns image $\boldsymbol{x}$, label y, and a gradient image $\boldsymbol{\nabla_x} F$ as inputs, which is trained to maximize the loss function of the classifier. In other words, $F$ and $G$ play the following two-player minimax game:

$$\mathbb{E}_{\boldsymbol{x}, \mathrm{y} \sim P_{\mathrm{data}}} \min_{\boldsymbol{\theta}} \max_{\boldsymbol{\phi}} \left[ \mathrm{Loss}(\boldsymbol{x} + G(\boldsymbol{\nabla_x} F, \boldsymbol{x}, \mathrm{y}; \boldsymbol{\phi}), \mathrm{y}; \boldsymbol{\theta}) - c_L \| G(\boldsymbol{\nabla_x} F, \boldsymbol{x}, \mathrm{y}; \boldsymbol{\phi}) \|_2^2 \right] \quad (4)$$

By the time this minimax game is complete, the classifier will have been trained with various attacks produced by the generator with the enhanced robustness against powerful adversarial attacks, while the generator will no longer find any vulnerability in the classifier, therefore only producing random noises. In Equation (4), $c_L$ is a hyper-parameter that adjusts the ratio between two cost functions. If $c_L$ is very low, it will only find trivial solutions with extremely large perturbation power, and if $c_L$

is very high, it will only generate zero-perturbation images. Therefore, it is crucial to determine an appropriate $c_L$. The theoretical meaning of $c_L$ will be discussed in Appendix A.

We believe that the strength of our algorithm is the flexibility of the generator's attack method. Conventional adversarial training algorithms such as FGM have a fixed attack method. For example, adversarial perturbation in FGM is generated by normalizing the gradient of a data sample, which is scaled by $\epsilon$. A classifier can be easily made robust against adversarial examples generated by these fixed methods. The following are two possible methods.

CASE 1. The classifier is trained to gradually reduce the gradient at data points. Gradually reducing the gradient prevents FGM from generating meaningful adversarial example from the gradient (also known as a gradient masking problems).

CASE 2. The classifier can only be trained to reduce the loss under the surface of the $\epsilon$-norm ball centered around a data point. As a result, the network would be robust against adversarial examples generated by FGM, but could still be very vulnerable against adversarial examples located inside the $\epsilon$-norm ball.

Through the above methods, a classifier can be easily overfit to FGM attacks, but it becomes more vulnerable to other powerful attacks such as Carlini&Wagner $L_2$ attacks as shown in Figure 2. Through our experiments under FGM adversarial training, we observed a CASE 1 overfitting problem when $\epsilon$ is small, and a CASE 2 overfitting problem when $\epsilon$ is large. The above gradient masking problem has also been discussed by other papers on adversarial training (Tramèr et al., 2018). This problem also occurs when the iteration number of PGD is relatively low. On the other hand, our algorithm continuously updates the parameters of a generator to maximize the classifier loss, which tends to produce more general attacks to a given classifier. This means that a classifier cannot easily overfit to attacks from a generator in our method. Experimental results also support our arguments.

## 4 EXPERIMENT

### 4.1 EXPERIMENTAL SETUP

We used CIFAR-10 and CIFAR-100 for our datasets, to verify the robustness of our trained network. We normalized the pixel value of the image to [0,1] prior to network training. This section only presents the results from CIFAR-100. The experiment with CIFAR-10 showed similar results, and interested readers can refer to the appendix for its results.

The model architecture and parameters for CIFAR are given in Table 1. We used conventional ConvPool-CNN as the classifier network, and the generator network was designed to efficiently use gradients, images, and labels. One might assume that hyperbolic tangent function should be used for the final activation function of the generator network. However, if a hyperbolic tangent function is used for generating the perturbation image, the adversarial image created must be clipped again to the proper value of the image, i.e. $0 <= x_i + \delta_i <= 1$ for all $i$. This is known as a box-constraint problem, which might cause the network to get stuck in extreme regions (Carlini & Wagner, 2017). Therefore, we practiced the following technique proposed by Carlini & Wagner (2017) to avoid the clipping issue.

$$\boldsymbol{x}_{adv} = \frac{1}{2}(\tanh(\tanh^{-1}(2\boldsymbol{x} - 1) + G(\boldsymbol{\nabla}_{\boldsymbol{x}}F, \boldsymbol{x}, \mathrm{y}; \boldsymbol{\phi})) + 1) \tag{5}$$

For our baseline for comparison, we used a naive network trained only with clean examples. For our control group, we set Goodfellow et al. (2016)'s adversarial training with Fast Gradient Method (FGM), and Madry et al. (2018)'s adversarial training with Projected Gradient Descent (PGD). For attack methods, we used FGM, Momentum Iterative Method (MIM), DeepFool, and Carlini&Wagner (C&W), and evaluated the robustness of the network through the accuracy of the adversarial examples and the mean and median value of the $L_2$ norm of the perturbaion generated by each attack. All the attacks and adversarial training methods above are $L_2$-bounded. Detailed evaluation method will be discussed in section 4.2.

All of our experiments used a single RTX 2080 ti GPU with Cleverhans adversarial examples library (Papernot et al., 2018) to construct adversarial attacks, build defenses, and make comparison more effectively.

Table 1: Model architectures and parameters for CIFAR datasets

| Classifier Network | Generator Network | Parameter | |
|---|---|---|---|
| Input: $x$ ($32 \times 32 \times 3$) | Input: $y$ (10 or 100) | Optimizer | Adam |
| $3 \times 3$ Conv 64 | Dense 1024 | Learning Rate | 0.0005 |
| $3 \times 3$ Conv 128 | Concatenate with $y$ | Batch Size | 128 |
| $2 \times 2$ AvgPool | Dense 8192 | Adv Coefficient | 1.0 |
| $3 \times 3$ Conv 128 | Reshape $8 \times 8 \times 128$ | PGD iter | 10 |
| $3 \times 3$ Conv 256 | Concatenate with $y$ (reshape) | Dropout | - |
| $2 \times 2$ AvgPool | $3 \times 3$ Upconv 128, stride=2 | Weight Decay | 0 |
| $3 \times 3$ Conv 256 | Concatenate with $y$ (reshape) | Ema decay | 0.998 |
| $3 \times 3$ Conv 512 | $3 \times 3$ Upconv 128, stride=2 | Max Epochs | 200 |
| $2 \times 2$ AvgPool | Concatenate with $[\boldsymbol{x}, \boldsymbol{\nabla_x} F]$ | FGS attack eps | 0.5 |
| $3 \times 3$ Conv 10 or 100 | $3 \times 3$ Upconv 128, stride=1 | MIM attack eps | 0.5 |
| GlobalAvgPool | $3 \times 3$ Upconv 3, stride =1 | MIM attack iter | 100 |
| Softmax | | DF attack iter | 100 |
| Output: 10 or 100 class probabilities | Output: $32 \times 32 \times 3$ perturbation | CW attack iter | 100 |

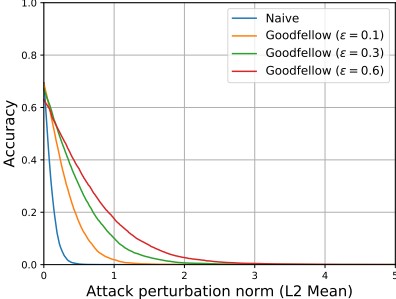 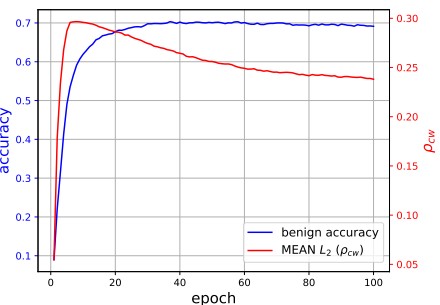

Figure 2: Trade-off relationship between benign accuracy and the adversarial robustness. Left: Perturbation power-accuracy graph for FGM adversarial training with various epsilon against CW attack. The bigger the epsilon, the more robust the network to adversarial attacks, but less benign accuracy. Right: The network with FGM adversarial training tend to overfit easily because of its fixed attack algorithm. As the training progresses, the benign accuracy rises, whereas the adversarial robustness declines.

## 4.2 EVALUATION METHOD

We applied the following average distortion metric suggested by Carlini & Wagner (2017), in order to fairly evaluate the robustness of the network for various adversarial attacks.

$$\rho := \frac{1}{|\mathcal{D}|} \sum_{\boldsymbol{x}_k \in \mathcal{D}} \|\Delta_{\boldsymbol{x}_k}\|_2, \quad \text{where } \mathcal{D} \text{ is a successful adversarial example set} \tag{6}$$

The above $\rho$ represents the mean value of $L_2$ norm of the perturbation derived from the successful adversarial examples from the attack, the same value as the area under the curve in Figure 2. Although $\rho$ can be measured for any attack methods, it is best to measure $\rho$ for the most powerful attack. Thus, we used $\rho_{\text{cw}}$ for Carlini&Wagner $L_2$ attack on all of our experiments as the evaluation metric for robustness of the network.

However, it is not sufficient to use only the above metric in evaluating the robustness of the adversarial training algorithm. In most adversarial training process, there are some hyperparameters which could adjust the trade-offs of the accuracy of benign examples and the adversarial examples ($\epsilon$ for FGM, PGD adversarial training, $c_L$ for our algorithm). The above $\rho_{\text{cw}}$ tends to increase as the accuracy of the benign examples decline, as it can be shown in Figure 2. In an extreme case, if the network classify almost all the images as a single class, benign accuracy (the accuracy of clean examples) would converge to 1% (for CIFAR-100), but the $\rho_{\text{cw}}$ would spike. This trade-off occurs during the training process as well. Figure 2 illustrates how benign accuracy increases, and $\rho_{\text{cw}}$

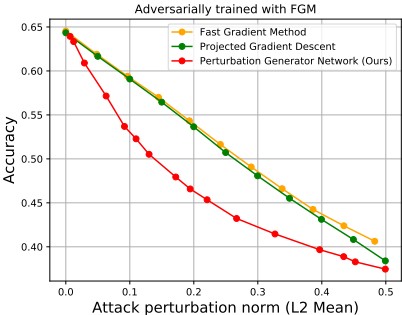 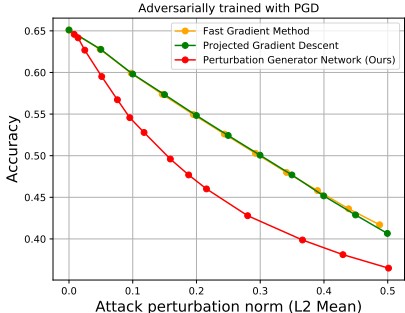

Figure 3: Evaluation of the attack performance of the proposed generator network. Accuracy vs perturbation norm was plotted for various attack methods. Left: a classifier was adversarially trained with Fast Gradient Method. Right: a classifier was adversarially trained with Projected Gradient Descent.

decreases during the training process. Thus, for fair comparison of the robustness of the networks, it is desirable to match the accuracy of the benign examples before comparing $\rho_{cw}$.

To better compare the performance of different adversarial algorithms, we must first train the models by adjusting hyper-parameters for each adversarial training, calculate benign accuracy and $\rho_{cw}$ for each trained model, and then draw a graph with the calculated values, connecting each relevant data point. Naturally, the structure of the classifier used for each adversarial training must be identical. We will call this graph robustness-curve. Figure 5 displays the robustness-curves for FGM adversarial training, PGD adversarial training, and our algorithm. It should be noted that the method with outer curve is a better adversarial training algorithm since $\rho_{cw}$ is higher at the same benign accuracy.

### 4.3 Attack performance of the perturbation generator network

In order to verify the attack performance of the proposed generator network, we used two fixed classifier networks adversarially trained with FGM ($\epsilon = 0.5$) and PGD ($\epsilon=0.5$), respectively. For each classifier network, we compared the performance of FGM attack, PGD attack, and PGN (proposed) attack. PGN was fully trained on the training dataset. We plotted the accuracy of the classifiers for these three attacks in terms of average perturbation distortion. The results are shown in Figure 3. For each model, it is observed that our algorithm is capable of much more powerful attack than PGD and FGM. This difference in performance clearly justifies the superior defense performance of the adversarial training with PGN in the following section.

The perturbation generator network can be trained to fully exploit incoming gradients, images, and labels to generate adversarial perturbation optimized for the classifier. As for FGM or PGD, they have a fixed norm, with a single attack direction towards the gradient. Now with PGN, it has no restriction on both direction and size, and can be transformed into an optimized perturbation for the classifier.

### 4.4 Defense performance on various attacks

Based on the methods introduced in 4.2, we compared the robustness of the network trained by our suggested technique with that trained by the conventional adversarial training methods. FGM attack, MIM attack, DeepFool, and Carlini&Wagner Attack were used as attack methods. In white-box attacks, adversarial examples were generated through direct access to the model's gradient, while in black-box attack, accuracy was measured through the adversarial examples produced by an independently trained network. Table 2 exhibits the robustness of the network when all the benign accuracy values of the adversarial networks are balanced to that of the naive network, and Figure 4 displays three perturbation $L_2$ power (x) - accuracy (y) graphs for C&W attack, with the benign accuracy for each adversarial networks set to 68%, 66%, and 63%, respectively.

FGM and MIM are attack methods that find the adversarial examples that can maximize the loss function of the classifier network on fixed $L_2$ norm of perturbation power, so the higher the accuracy

Table 2: The comparison of the performance of the conventional adversarial training algorithms and our algorithm with $\epsilon = 0.02$ and $c_L = 50$. Benign accuracy of all defenses were balanced out with that of the baseline network before the comparison. Column 3, 6: Prediction accuracies of White-Box attack and Black-Box attack for each attack algorithms. Column 4: MEAN $L_2$ norm of the adversarial perturbation ($\rho$, which is defined in Equation (6)). Column 5: Median $L_2$ norm of the adversarial perturbations.

| DEFENSE | ATTACK | ACCURACY W-BOX | MEAN $L_2$ W-BOX | MEDIAN $L_2$ W-BOX | ACCURACY B-BOX | BENIGN ACCURACY | TRAINING TIME (SEC/EPOCH) |
|---|---|---|---|---|---|---|---|
| Baseline | FGM | 0.1173 | - | - | 0.3843 | | |
| | MIM | 0.0167 | - | - | 0.2913 | | |
| | Deepfool | - | 0.0994 | 0.0626 | 0.6688 | 0.7002 | 8.18 |
| | C&W | - | 0.0791 | 0.0503 | 0.6659 | | |
| Goodfellow et al. (2015) | FGM | 0.2213 | - | - | 0.5211 | | |
| | MIM | 0.1068 | - | - | 0.5115 | | |
| | Deepfool | - | 0.1669 | 0.1089 | 0.6902 | 0.6993 | 25.1248 |
| | C&W | - | 0.1334 | 0.0867 | 0.6894 | | |
| Madry et al. (2018) | FGM | 0.2139 | - | - | 0.5164 | | |
| | MIM | 0.1000 | - | - | 0.5014 | | |
| | Deepfool | - | 0.1625 | 0.1061 | 0.6906 | 0.7000 | 175.8571 |
| | C&W | - | 0.1305 | 0.0851 | 0.6880 | | |
| Ours | FGM | 0.3906 | - | - | 0.6428 | | |
| | MIM | 0.3444 | - | - | 0.6456 | | |
| | Deepfool | - | 0.3184 | 0.2064 | 0.6958 | 0.7004 | 51.7681 |
| | C&W | - | 0.2617 | 0.1674 | 0.6961 | | |

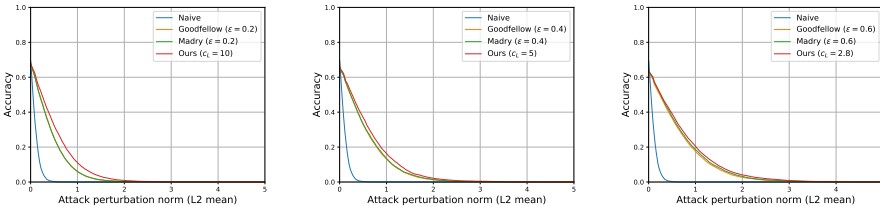

Figure 4: The comparison of the robustness of the defense methods with various benign accuracy. To properly compare the robustness of the networks, the benign accuracy of the networks needs to be balanced out. The graph displays three different curves, each representing the accuracy of the perturbation power with respect to the perturbation norm of Carlini&Wagner $L_2$ attack with different benign accuracy of 68%, 66%, and 63%, respectively.

of the adversarial examples, the more robust the network. On the other hand, DeepFool and C&W attack find the adversarial examples with the lowest $L_2$ norm of perturbation power that can fool the network; therefore, the robust network would have higher mean and median values of the adversarial perturbation power.

As you can see from Table 2, our algorithm outperforms the other adversarial training algorithms against all the attack methods. In white-box attacks, our algorithm showed the highest accuracy of adversarial examples against FGM and MIM attacks, and the highest power of adversarial perturbations by DeepFool and Carlini&Wagner. Also, in black-box attacks, our method proved to classify the adversarial examples with greater accuracy compared with the other adversarial training algorithms. According to Table 2, FGM adversarial training and PGD adversarial training show a very similar performance. This is because minimal $\epsilon$ was applied to match the benign accuracy of the baseline network. Since a neural network is locally linear, this minimal $\epsilon$ would make PGD and FGM generated adversarial examples to be almost identical. As you can see from Figure 4 and Figure 5, as $\epsilon$ increases, Madry's method shows a more robust performance compared with Goodfellow's method.

Training speed is also a crucial issue in adversarial training. The proposed algorithm is slower than FGM because it trains the generator after finding the gradient image, while FGM immediately uses the gradient image to train the classifier. On the other hand, our algorithm is faster than Madry's which use PGD (multi-step gradient descent) to find the adversarial image. Note that the more iteration steps of PGD, the larger the speed-gap between Madry's and our algorithm we get.

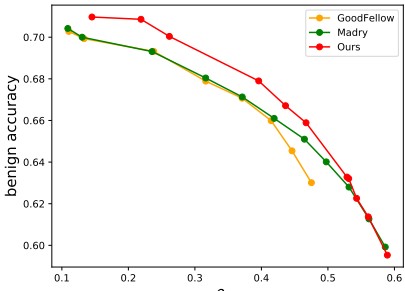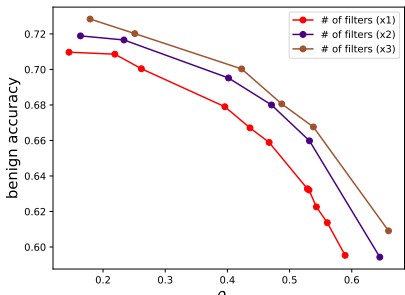

Figure 5: Robustness-curve: A plot showing the relationship between benign accuracy and $\rho_{\mathrm{cw}}$ by changing the hyper-parameters of each adversarial training algorithm. Left: For FGM and PGD adversarial training, each data point was acquired through changing $\epsilon$, whereas for our algorithm, each data point was acquired through changing $c_L$. The outer curves are considered more robust adversarial algorithms. Right: Robustness-curves for our algorithm under different capacities of the classifier. It shows that the classifier is still underfitting in terms of adversarial robustness.

### 4.5 VARYING HYPER-PARAMETERS

As mentioned in Section 4.2, adversarial training algorithms have a trade-off relationship between benign accuracy and robustness metric $\rho_{\mathrm{cw}}$. Figure 5 visualizes the relationship with a plot consisting of the data points of benign accuracy and $\rho_{\mathrm{cw}}$, which are collected by using various hyper-parameters for each adversarial training. For FGM and PGD adversarial training, the data points with higher $\rho_{\mathrm{cw}}$ are models trained with bigger $\epsilon$, while for our algorithm, the data points with larger $\rho_{\mathrm{cw}}$ are models trained with lower $c_L$. It should be noted that the robustness-curve is an appropriate indicator for performance evaluation of the adversarial training algorithms, since it displays a comprehensive set of $\rho_{\mathrm{cw}}$ with corresponding benign accuracy. As demonstrated in Figure 5, our algorithm outperforms all the other adversarial training algorithms under all benign accuracy.

Furthermore, we plotted the robustness-curve by proportionally increasing the number of filters in each convolutional layer of the classifier. As can be observed in the second plot of Figure 5, the robustness-curve moves to the right as the capacity of the model increases, which means that the classifier may still be underfitted. In other words, the classifier trained with only clean examples tend to overfit easily to the training data with even a low capacity, whereas the classifier trained with various adversarial examples tend to underfit instead even with higher capacity networks. Although we were not able to deal with higher capacity due to the limits of the current hardware technology, it is expected that a far greater network capacity may be needed to achieve a human-level robustness.

## 5 CONCLUSION

This study proposed a novel adversarial training method that boosts the robustness of a deep neural network against adversarial attacks. Based on a GAN framework, the classifier network and the generator network play a two-player minimax game, which improves the robustness of a classifier against adversarial examples. In generating adversarial examples, we use a trainable perturbation generator network instead of a fixed function as in most of conventional adversarial training methods. this method tend to overfit less, and strengthens the robustness against many different kinds of attacks. Our proposed method is far more robust than existing adversarial training techniques. Since it computes adversarial examples through one-step inference, it is also more advantageous in training speed, compared to other techniques that use multiple steps in inner maximization.

Our experiment with CIFAR datasets have also proved the advantage of our approach, as the network trained by our method showed improved robustness and the state-of-the-art performance against various attacks with different noise power. Although the proposed approach compares favorably with other methods, it is believed that there is still room for improvement. One future direction is to study a generator network which is most effective for adversarial training.

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

APPENDIX

# A THEORETICAL BACKGROUND

Madry et al. (2018) has presented the following adversarial loss instead of the conventional loss in his paper.

$$\rho_{\text{madry}}(\boldsymbol{\theta}) = \mathbb{E}_{(\boldsymbol{x},\text{y})\sim P_{\text{data}}} \max_{\boldsymbol{\delta}\in\mathbb{S}} [\text{Loss}(\boldsymbol{x}+\boldsymbol{\delta},\text{y};\boldsymbol{\theta})] = \mathbb{E}_{(\boldsymbol{x},\text{y})\sim P_{\text{data}}} \max_{\boldsymbol{\delta}\in\mathbb{S}} \left[\log\left(\frac{1}{F(\boldsymbol{x}+\boldsymbol{\delta};\boldsymbol{\theta})_{\text{y}}}\right)\right] \quad (7)$$

This signifies that among the given training data points, it finds the data point that has the maximum loss against perturbation from the allowed perturbations set $\mathbb{S}$, and minimizes that specific loss. In other words, it trains Classifier $F$ to satisfy $F(\boldsymbol{x}+\boldsymbol{\delta};\boldsymbol{\theta})_{\text{y}} = 1$ for all $\boldsymbol{\delta}$ in $\mathbb{S}$ in allowed perturbations set $\mathbb{S}$, which is only feasible when $\mathbb{S}$ is a very small norm ball. However, since our perturbation generator network can create perturbations with any size of power, merely applying the above adversarial loss would generate only the trivial perturbations with extremely high power. In order to extend the allowed perturbations set $\mathbb{S}$ to all possible perturbations, we assume that optimal $F(\boldsymbol{x}+\boldsymbol{\delta};\boldsymbol{\theta})_{\text{y}}$ has a normal distribution over $L_2$ norm of the $\boldsymbol{\delta}$ as shown in Figure 6. To be more precise, we want to train Classifier $F$ to satisfy $F(\boldsymbol{x}+\boldsymbol{\delta};\boldsymbol{\theta})_{\text{y}} = p_{\text{n}}(\|\boldsymbol{\delta}\|_2)$, where $\text{n} \sim \mathcal{N}(0,\sigma^2)$ and $p_{\text{n}}$ is the Gaussian distribution function with 0 mean and $\sigma^2$ variance, and our adversarial loss corresponding to Equation (7) is as follows:

$$\begin{aligned}\rho_{\text{ours}}(\boldsymbol{\theta}) &= \mathbb{E}_{\boldsymbol{x},\text{y}\sim P_{\text{data}}} \max_{\boldsymbol{\delta}\in\mathbb{S}} \left[\log\frac{p_n(\|\boldsymbol{\delta}\|_2)}{F(\boldsymbol{x}+\boldsymbol{\delta};\boldsymbol{\theta})_{\text{y}}}\right]\\ &= \mathbb{E}_{\boldsymbol{x},\text{y}\sim P_{\text{data}}} \max_{\boldsymbol{\delta}\in\mathbb{S}} \left[\log\frac{e^{-\frac{1}{2\sigma^2}\|\boldsymbol{\delta}\|_2^2}}{\sqrt{2\pi}\sigma} - \log F(\boldsymbol{x}+\boldsymbol{\delta};\boldsymbol{\theta})_{\text{y}}\right]\\ &= \mathbb{E}_{\boldsymbol{x},\text{y}\sim P_{\text{data}}} \max_{\boldsymbol{\delta}\in\mathbb{S}} \left[-\frac{1}{2\sigma}\|\boldsymbol{\delta}\|_2^2 - \log F(\boldsymbol{x}+\boldsymbol{\delta};\boldsymbol{\theta})_{\text{y}} - \log\sqrt{2\pi}\sigma\right]\\ &= \mathbb{E}_{\boldsymbol{x},\text{y}\sim P_{\text{data}}} \max_{\boldsymbol{\delta}\in\mathbb{S}} \left[\text{Loss}(\boldsymbol{x}+\boldsymbol{\delta},\text{y};\boldsymbol{\theta}) - c_L\|\boldsymbol{\delta}\|_2^2 - C\right]\end{aligned} \quad (8)$$

Suppose $\delta^*$ is defined as $\delta^*(F,\boldsymbol{x},\text{y}) = \underset{\boldsymbol{\delta}}{\text{argmax}} \left[\text{Loss}(\boldsymbol{x}+\boldsymbol{\delta},\text{y};\boldsymbol{\theta}) - c_L\|\boldsymbol{\delta}\|_2^2 - C\right]$. then,

$$\begin{aligned}\min_{\boldsymbol{\theta}} \rho_{\text{ours}}(\boldsymbol{\theta}) &= \mathbb{E}_{\boldsymbol{x},\text{y}\sim P_{\text{data}}} \min_{\boldsymbol{\theta}} \left[\text{Loss}(\boldsymbol{x}+\delta^*(F,\boldsymbol{x},\text{y}),\text{y};\boldsymbol{\theta}) - c_L\|\delta^*(F,\boldsymbol{x},\text{y})\|_2^2 - C\right]\\ &\approx \mathbb{E}_{\boldsymbol{x},\text{y}\sim P_{\text{data}}} \min_{\boldsymbol{\theta}} \max_{\boldsymbol{\phi}} \left[\text{Loss}(\boldsymbol{x}+G(\boldsymbol{\nabla}_{\boldsymbol{x}}F,\boldsymbol{x},\text{y};\boldsymbol{\phi}),\text{y};\boldsymbol{\theta}) - c_L\|G(\boldsymbol{\nabla}_{\boldsymbol{x}}F,\boldsymbol{x},\text{y};\boldsymbol{\phi})\|_2^2 - C\right]\end{aligned}$$
$$(9)$$

$G$ would converge to $\delta^*$, assuming $G$ has sufficiently high capacity and $\boldsymbol{\nabla}_{\boldsymbol{x}}F$ provides enough information on the classifier $F$, and the constant value $C$ could be ignored since we are only interested in finding the parameter $\boldsymbol{\theta}$ of the classifier. We can derive the Equation (4), with $c_L = \frac{1}{2\sigma}$. The optimal point for $F$ and $G$ would be the point where $\text{Loss}(\boldsymbol{x}+\boldsymbol{\delta},\text{y};\boldsymbol{\theta}) \leq c_L\|\boldsymbol{\delta}\|_2^2 + C$ in all data points, and here we can find the classifier network $F$ that has the improved robustness against adversarial examples.

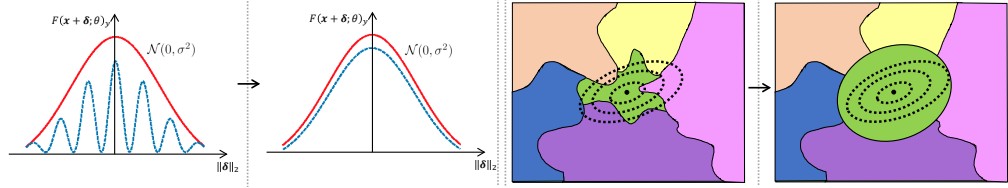

Figure 6: Adversarial Robustness with Gaussian Normal Distribution. Left: The red curve indicates our target Gaussian normal probability function with mean 0 and variance $\sigma^2$. The blue dotted curve indicates the classifiers class probability of the label in accordance with the $L_2$ norm of the perturbation. As the training progresses, the classifiers class probability converges to the target function, ensuring the robustness of the network. Right: Conceptual illustration of the adversarial training. By minimizing the loss function on the region with large adversarial loss, the network becomes increasingly robust against adversarial attacks.

## B  PERFORMANCE FOR CIFAR-10

Table 3: The comparison of the performance of the conventional adversarial training algorithms and our algorithm with $\epsilon = 0.1$ and $c_L = 50$.

| DEFENSE | ATTACK | ACCURACY W-BOX | MEAN $L_2$ W-BOX | MEDIAN $L_2$ W-BOX | ACCURACY B-BOX | BENIGN ACCURACY | TRAINING TIME (SEC/EPOCH) |
|---|---|---|---|---|---|---|---|
| Baseline | FGM | 0.2934 | - | - | 0.4979 | | |
| | MIM | 0.0939 | - | - | 0.3633 | | |
| | Deepfool | - | 0.2129 | 0.1844 | 0.8527 | 0.9189 | 8.28 |
| | C&W | - | 0.1742 | 0.1596 | 0.8611 | | |
| Goodfellow et al. (2015) | FGM | 0.6335 | - | - | 0.8428 | | |
| | MIM | 0.5543 | - | - | 0.851 | | |
| | Deepfool | - | 0.5135 | 0.4502 | 0.9119 | 0.918 | 25.1424 |
| | C&W | - | 0.4196 | 0.3919 | 0.9116 | | |
| Madry et al. (2018) | FGM | 0.6259 | - | - | 0.8369 | | |
| | MIM | 0.5426 | - | - | 0.8468 | | |
| | Deepfool | - | 0.4977 | 0.437 | 0.9104 | 0.9172 | 175.2541 |
| | C&W | - | 0.4091 | 0.382 | 0.91 | | |
| Ours | FGM | 0.6534 | - | - | 0.8501 | | |
| | MIM | 0.5958 | - | - | 0.8568 | | |
| | Deepfool | - | 0.5102 | 0.4483 | 0.9127 | 0.9186 | 52.1851 |
| | C&W | - | 0.4365 | 0.4062 | 0.912 | | |

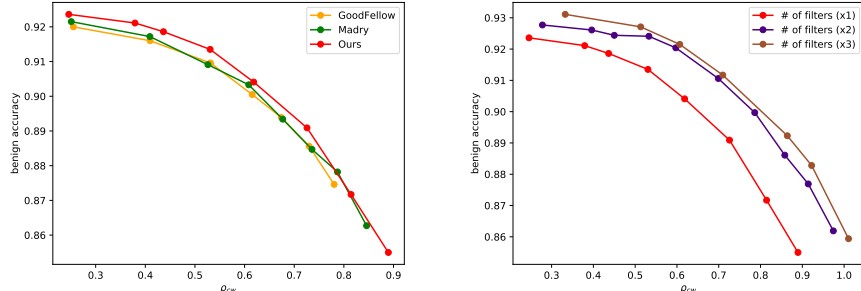

Figure 7: Robustness-curve. Left: Varying hyper-parameter, Right: Varying capacity

# C SOME ADVERSARIAL EXAMPLES GENERATED BY THE PROPOSED GENERATOR NETWORK

Table 4: Adversarial examples for various perturbation powers from the CIFAR-100 dataset. These images are generated from the generator which is fully trained to maximize the classification loss. Labels denote corresponding classification results.

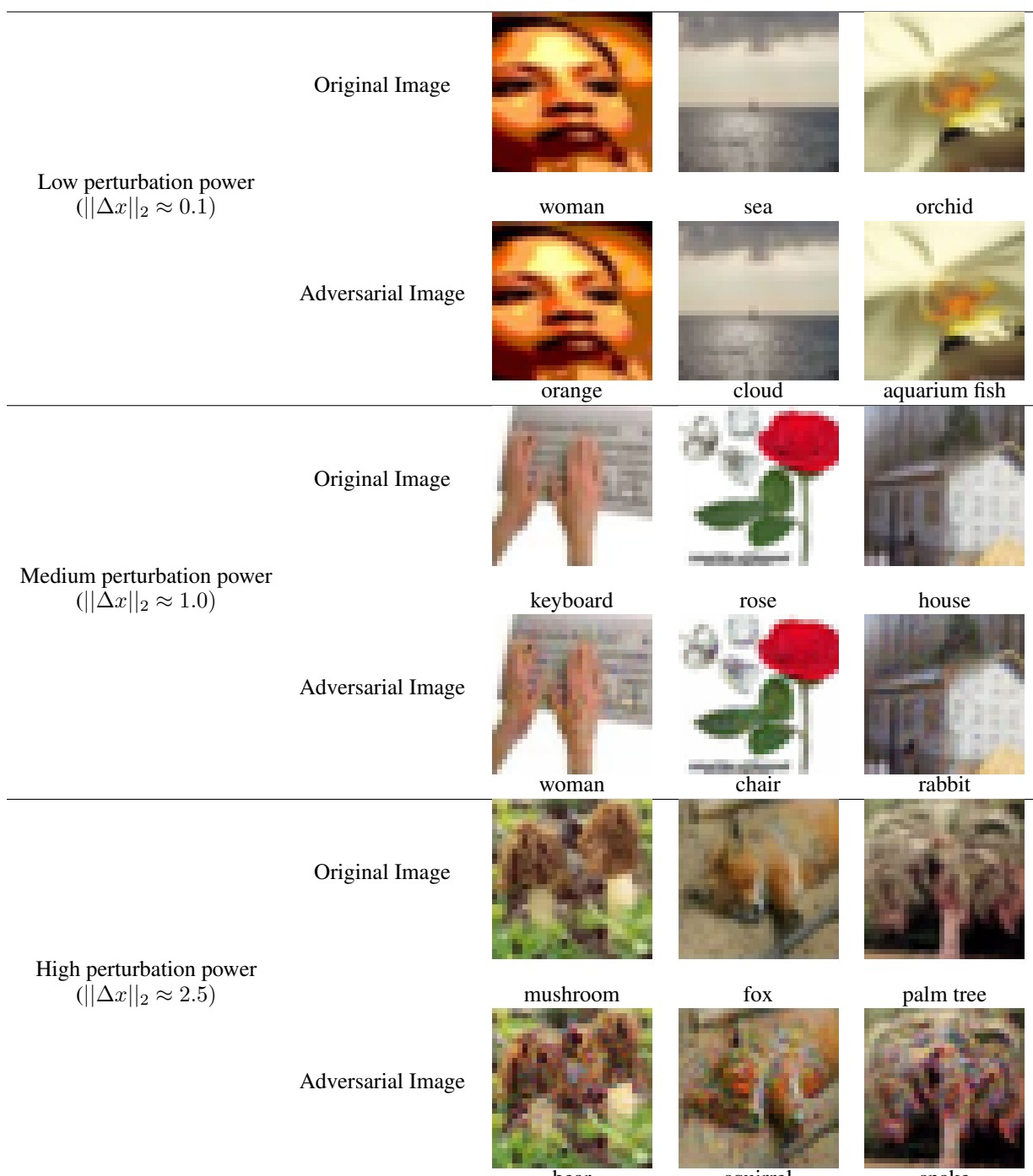

## D EFFECT OF EACH INPUT ELEMENT ON THE ROBUSTNESS OF THE NETWORK:ABLATION STUDY

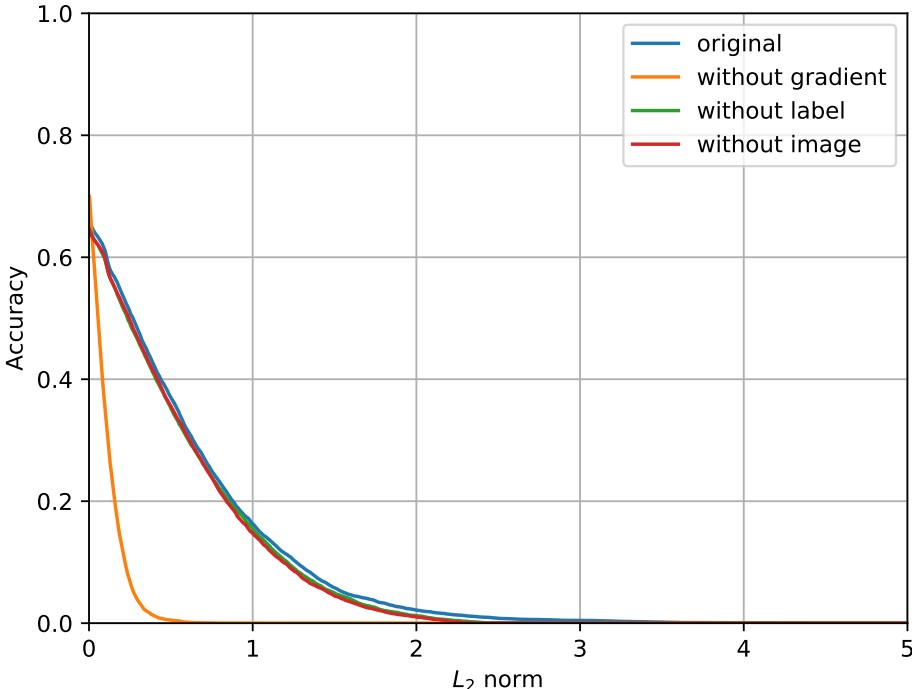

Figure 8: In order to measure how much contribution each input make in terms of the robustness of the classifier, we removed each input one by one, and plotted an accuracy vs average distortion curve. It can be observed that the gradient is the most important factor in classifier robustness, and the other two factors have relatively smaller impact on classifier robustness.

