# OpenReview forum: "Adversarial training with perturbation generator networks"
_ICLR.cc/2020/Conference — Reject_

### Official Review · AnonReviewer1 · 2019-10-20
**Official Blind Review #1**

**Rating:** 3

**Review:**

This paper introduces a new adversarial training approach, where a generator is used to generate the most challenging adversarial examples and the classifier is trained to correctly classify the generated adversarial examples. In this way, the robustness of the classifier is expected to be improved. For the generator, the input includes the original data sample, the gradient of the classifier, and the true label of the data sample. The paper is clearly presented and easy to follow. The experimental results relatively support the main claims of the paper.

The reasons that I go towards mild rejection for this paper are as follows:

1. The general idea of improving the robustness of a classifier to by feeding adversarial examples to it is not a new idea. As shown in the paper, the idea has been adopted in Goodfellow et al. (2015) and Madry et al. (2018). In addition, [1] comprehensively studied how to ensemble different adversarial attacks as "training data" of a classifier to improve its robustness. The paper generates adversarial examples with a generator that takes various things as inputs, which can be viewed as an extension to the method in [2]. [2] introduced a generator that only takes the data sample as inputs and the difference to the paper is that the one in this paper additionally takes gradients and label as inputs. With Goodfellow et al. (2015), Madry et al. (2018), and [1], it may not be too hard to apply the method in [2] to improve classifier robustness. Therefore, it is questionable whether the additional information used in the generator of this paper helps and how it helps. I did not see any comparison in the experiments.

2. Although the experiments look to be well conducted, it is not comprehensive enough. For the defence methods in comparison, only two approaches that fall into the exact same line of the proposed method are included. In this line, I think it is necessary to compare with [1], which combines various attacks. Moreover, I would also expect a comparison with other closely related methods such as [3] and [4], to further demonstrate the effectiveness of the proposed one.

3. It is also suggested to do ablation tests on the inputs of the generator to see how each part of the inputs helps. If gradients and true label are removed from the generator, it reduces to [2]. Therefore, those ablation tests also help the comparison with [2].

Minor comments:

1. It could be interesting to visualise the perturbations that are generated from the generator.

2. Some of the figures and captions are too small to see in a printout.


[1] Tramèr, Florian, Alexey Kurakin, Nicolas Papernot, Ian Goodfellow, Dan Boneh, and Patrick McDaniel. "Ensemble Adversarial Training: Attacks and Defenses." (2018).

[2] Xiao, Chaowei, Bo Li, Jun-Yan Zhu, Warren He, Mingyan Liu, and Dawn Song. "Generating adversarial examples with adversarial networks." arXiv preprint arXiv:1801.02610 (2018).

[3] Samangouei, Pouya, Maya Kabkab, and Rama Chellappa. "Defense-GAN: Protecting Classifiers Against Adversarial Attacks Using Generative Models." (2018).

[4] Matyasko, Alexander, and Lap-Pui Chau. "Improved network robustness with adversary critic." In Advances in Neural Information Processing Systems, pp. 10578-10587. 2018.

**Experience Assessment:**

I have published one or two papers in this area.

**Review Assessment: Checking Correctness Of Derivations And Theory:**

I carefully checked the derivations and theory.

**Review Assessment: Checking Correctness Of Experiments:**

I assessed the sensibility of the experiments.

**Review Assessment: Thoroughness In Paper Reading:**

I read the paper at least twice and used my best judgement in assessing the paper.

---

> ### Author Response · Authors · 2019-11-11
> **Response to Reviewer #1**
>
> I would like to thank you for your valuable review. Here are some initial replies on your questions.
>
> 1.	First, you mentioned Goodfellow’s and Madry’s can just as easily improve the robustness of the classifier by feeding adversarial examples. However, what is being claimed is that the previous methods like Goodfellow’s and Madry’s have certain drawbacks, and our technique can mitigate these drawbacks. Our algorithm is more robust than Goodfellow’s, and it is faster than Madry’s. Thus, we can train on large-scale datasets like Imagenet, which was almost infeasible with the PGD methods.
> Although we agree that [2] and our technique have a similarity in that they both uses a neural network in creating adversarial examples, but I would like to emphasize that our method is different in three ways. First, we only used GAN’s “mini-max framework,” whereas [2] used GAN itself. As can be seen in our proof, we do not have a discriminator network. Second, the main purpose of the method of [2] is to attack the network, whereas the goal of our method is to make the classifier robust through iterative training process. Third, as you have mentioned, our method additionally takes gradients and label as inputs. In adversarial training, it is essential to take gradients as input in order to adaptively attack the classifier. Our experiment results show that gradients is the most critical factor in improving the robustness of a classifier, and the next important factors are ordered as the label and the image data.
>
> 2.	Thank you for introducing many interesting methods. However, we believe that methods introduced in [1] and [3] are not suitable for comparison for the following reasons. First, [1] studies an ensemble method in which adversarial examples are collected from various network model, and is used to make the network robust. Since this method uses multiple networks to improve robustness, it is difficult to compare with algorithms with a single network. Practically, any algorithms including ours can enhance its performance through an ensemble method. Regarding the method introduced in [3], it does not make the classifier robust, and the classifier is not strengthened in the method. Instead, the adversarial examples are transformed into clean examples with GAN before they are fed into the classifier. This is not an adversarial training method, so it is not suitable to compare it with our algorithm. Unlike the methods of [1] and [3], we believe that method [4] is a good candidate for comparison, and we will carefully review it and try to compare with our algorithm.
>
> 3.	As mentioned above, our experiment shows that the gradient is the most important factor in classifier robustness, with the second and the third being the label and the image, respectively. We will add the results of ablation tests on each input, and explain how it affects the classifier robustness in Appendix
>
> Thank you for pointing out some minor issues. We will add visualized image of the perturbation in Appendix, and we will also enlarge the size of the figure and caption.
>
> We appreciate your kind review again, and we are currently revising the manuscript based on what you pointed out.

---

### Official Review · AnonReviewer2 · 2019-10-23
**Official Blind Review #2**

**Rating:** 3

**Review:**

This paper proposes to use the GAN framework for adversarial training. The proposed algorithm is mini-max loss plus L2-regularization on the perturbations generated by a generator network. Additionally, the paper shows a mathematical interpretation of the L2-term. Experiments on CIFAR-10 and CIFAR-100 shows that the algorithm achieved higher clean/adversarial accuracy in the settings.

I vote for rejection. A major difference between the proposed method and adversarial training variants is whether we use neural networks or gradient-based algorithms to calculate the internal maximum in the mini-max formulation of the robust training. The focus of discussions in this paper should be why the former performs better. However, this paper does not provide enough justifications. Experiments are not convincing to support its advantage.

Major comments:
(1) On page 2, this paper claims, "The gradient descent based adversarial examples for robust optimization is not adaptive. Therefore, those neural networks are vulnerable to other types of adversarial attacks (Athalye et al., 2018)." Please expand these two sentences.
(2) It is weird to observe that adversarial training with PGD performs worse than FGM. It is possible that the baselines are weaker than they should be. For example, if evaluations use L2-based adversarial training as baselines, it can be improper. Please refer to this comment on OpenReview: https://openreview.net/forum?id=Hk6kPgZA-&noteId=BJVnpJPXM .

======= Update =======

Thank you for the replies and updates. After reading the comments and the updated draft, I am still not convinced that the proposed method should be superior to existing attacks. Even though I agree that the proposed method might outperform PGDs or other attacks in some settings, the paper needs to confirm it either theoretically or empirically. However, it was doubtful whether the baseline methods are strong enough. The concern remained after authors' responses. Hence I keep my score.

**Experience Assessment:**

I have published one or two papers in this area.

**Review Assessment: Checking Correctness Of Derivations And Theory:**

I assessed the sensibility of the derivations and theory.

**Review Assessment: Checking Correctness Of Experiments:**

I assessed the sensibility of the experiments.

**Review Assessment: Thoroughness In Paper Reading:**

I read the paper at least twice and used my best judgement in assessing the paper.

---

> ### Author Response · Authors · 2019-11-11
> **Response to Reviewer #2**
>
> I would like to thank you for your valuable review. We admit that we did not elaborate enough why our algorithm outperforms other conventional algorithms. So, we will explain further why our algorithm is better by giving you various examples.
>
> First of all, it should be noted that the conventional adversarial training algorithms have a ‘fixed’ method in generating adversarial examples. For example, in FGM, adversarial perturbation is generated by normalizing the gradient of the data sample, and then multiplying by the $\epsilon$. A classifier can be easily made robust against adversarial examples generated by these kind of fixed methods. The following are two possible scenario.
>
> CASE 1. The classifier is trained to gradually reduce the gradient at the data points. Gradually reducing the gradient prevents FGM from generating meaningful adversarial example from the gradient. This is also called a gradient masking problem.
>
> CASE 2. The classifier can only be trained to reduce the loss under the surface of the $\epsilon$-norm ball centered around the data point. The network would be robust against adversarial examples generated by FGM, but could still be very vulnerable against adversarial examples located inside the $\epsilon$-norm ball.
>
> We have observed from our experiment that under FGM adversarial training, we observed the CASE 1 overfitting problem when $\epsilon$ is small, and the CASE 2 overfitting problem when $\epsilon$ is large. Also, the gradient masking problem such as CASE 1 has been discussed by other papers on adversarial training [1][2]. Our algorithm continuously updates the parameter of the generator to maximize the classifier loss, which produces more general (adaptive) attacks (not restricted to a particular attack method) to the classifier. This means that the classifier may not get easily overfitted against adversarial attacks.
>
> Major comment 1): We will add additional explanation on the two sentences, with related experimental results in Section 4 in the revision.
>
> Major comment 2): Minimal $\epsilon$ was applied to match the benign accuracy of the baseline network. Since a neural network is locally linear, this minimal $\epsilon$ would make PGD and FGM generated adversarial examples be almost identical. As can  be seen in Figure 4 and Figure 5, as $\epsilon$ increases, PGD shows a more robust performance compared with the FGM method.
>
> We appreciate your kind review again, and we are currently revising the manuscript based on what you pointed out.
>
> [1] Tramèr, Florian, Alexey Kurakin, Nicolas Papernot, Ian Goodfellow, Dan Boneh, and Patrick McDaniel. "Ensemble Adversarial Training: Attacks and Defenses." (2018).
> [2] Anish Athalye, Nicholas Carlini, and David Wagner. “Obfuscated gradients give a false sense of security: Circumventing defenses to adversarial examples” (2018)

---

> > ### Comment · AnonReviewer2 · 2019-11-13
> > **Response**
> >
> > Thank you for your response.
> >
> > Comments to CASE 1 and CASE 2:
> >
> > My concern remains, and I am not sure whether the proposed algorithm can create more effective adversarial examples during the training phases.
> >
> > (1) To my understanding, CASE 1 is specific to FGM. PGD in Madry et al. starts gradient updates after adding random noises, which prevents the gradient masking to some extent (Tramèr et al.). Additionally, in the evaluation by Athalye et al., PGD-trained networks did not show the obfuscated gradients problem.
> > (2) CASE 2 is also specific to FGM. Moreover, we can prevent CASE 2 by randomly choosing $\epsilon$ in adversarial training using FGM.
> >
> > I encourage authors to demonstrate why or when classifiers overfit to PGD or other gradient-based adversarial attacks and do not overfit to PGN either theoretically or experimentally.
> >
> > Comments to other responses:
> >
> > Major comment 1): I will wait for the update.
> > Major comment 2): How about the point that L2-based PGD might perform worse compared to L\infty-based PGD when they defend networks from L2-based adversarial attacks (according to its authors)? If I can be sure that PGN performs better than existing most robust defense methods, my score might be raised to weak accept even without appropriate justifications. However, the experiments at the current stage might not provide appropriate comparisons with strong baselines.

---

> > > ### Author Response · Authors · 2019-11-14
> > > **Additional response**
> > >
> > > It appears that the number of PGD iterations that the reviewer and we assume is different. If the iteration number of the PGD is not sufficient, overfitting such as CASE 1 and CASE 2 may occur even in the case of PGD. We would like to emphasize that our algorithm outperforms PGD only when the iteration number of PGD is relatively low (in our experiments, the number of iterations of PGD is fixed to 10). If the number of iterations of PGD is large enough, the solution will be close to optimal, and it becomes the most powerful attack method among all the considered methods under such a condition. But, the problem is that the adversarial training time of PGD is proportional to the number of iterations, which makes it difficult to use practically for a large-scale dataset such as ImageNet (it may take several weeks or months to train). It should be noted that another strong advantage of the proposed method is its speed.
> > >
> > > Our proposed method is about 2 times slower than the FGM method because it alternately trains a generator and a classifier, but about 3 times faster than PGD (please refer to Table 1). The more PGD iterations, the bigger the gap of speed will be.
> > > We have experimentally demonstrated that the adversarially trained network with our method is more robust than those of the FGM and PGD. Our new experiments on the attack performance of a generator network also demonstrates the effectiveness of our method (these results are anonymously available on our github page https://github.com/ATPGN/ATPGN, and will soon be added to the revision). In other words, our method can be considered as an algorithm that combines the advantages of both FGM and PGD, and can be used practically even for large-scale datasets such as Imagenet.
> > >
> > > Major comment 2) Unfortunately, we did not conduct an experiment with $L_\infty$-based PGD because we naturally thought that $L_2$-based PGD would be more robust in case of $L_2$-based adversarial attack (I'm not sure that the claim is true).
> > > Apart from that, our algorithm can be easily extended to $L_\infty$-based PGN by removing the $L_2$ penalty term of the generator’s loss function, and using the hyperbolic tangent function (multiply with $\epsilon$) as an activation function in the generator’s output layer. The comparison of $L_\infty$-based PGN with $L_\infty$-based FGM and PGD would be a great future work, and it is believed that it is beyond the scope of this paper.

---

### Official Review · AnonReviewer3 · 2019-10-25
**Official Blind Review #3**

**Rating:** 3

**Review:**

The paper proposes to construct adversarial attacks by training a neural network to produce a distortion, rather than by constructing the distortion directly via PGD. It then uses this during adversarial training to produce networks that it purports to be more robust.

Pros:
-Faster speed of adversarial training could substantially democratize ability to work on adversarial ML
-Neural net attack could overcome gradient masking in some situations

Cons:
-Some of the main arguments seem incorrect, although the paper may still be strong if these incorrect arguments are removed and the experiments are further substantiated.

Details:
I think there are some interesting ideas here, which if further substantiated could be the base of a strong paper. However, I found the overall argument in the current paper to be misleading or at least mistaken. A core claim is that the neural-net generated attack is better than PGD because PGD only generates a "single attack" while the neural network generates many different attakcs. But PGD is an adaptive attack method that depends on the current model parameters, so it changes over the course of training just like the neural net attacker. Furthermore, both models approximate the same maximization over the L2 ball (in the neural net's case, rather than maximizing over the L2 ball we penalize the L2 norm, but a similar modification can easily be made to PGD). So it is not clear why we should expect the neural net to perform better, except perhaps that it is a different type of way of approximating the same objective that PGD approximates, and perhaps this approximation has some favorable properties? But this is never justified.

Here is one attempt to justify why the neural net attack should be better: perhaps while PGD often works well, it gets stuck on certain examples, e.g. due to gradient masking issues. While the neural net might initially get stuck on those examples (it also has to work by taking gradients), since it can share knowledge across examples it initially learns a good direction for the "stuck" examples by generalizing from the "unstuck" examples. This overall allows it to generate a more consistently good attack even in the presence of (partial) gradient masking. If we believe this story, then actually the neural net attacker should already work well just as an attack (i.e. just train it to generate good attacks against a fixed model for some set of examples). I would find this quite interesting if true but unfortunately it wasn't explored in the paper. One way to do it would be to take models that are believed to be robust based on attacking them with PGD, and then show that the neural net attacker is substantially more accurate.

I do find the point about faster speed interesting. This is given as a minor comment but it may be the biggest benefit of the current method. Adversarial training on large-scale datasets like ImageNet is computationally infeasible, and your method could potentially address this and allow people who don't have $100k+ compute budgets to actually work on adversarial ML.

I also have some comments on the evaluation:

-Reporting average accuracy across attacks in the evaluation is an inappropriate summary statistics. Min accuracy would be better, which in this case is zero for all methods. This suggests that you either did not adversarially train appropriately, or used too large of a norm (what norm did you use anyways? Or is this allowing the norm to be unbounded until you fool the model? In that case all the numbers should be zero for any reasonable attack.)

-For evaluation I would rather see the full curve of accuracy vs. allowed attack norm, rather than just a single summary metric. It's too hard to tell what is going on from a single number. It is also hard to compare against adversarial training methods that penalize the L2 norm versus constrain the L2 norm, as the former might do better than the latter simply due to more closely conforming to your evaluation of average distortion. Having the full curve helps better assess this. Average distortion is also only meaningful if the method is finding the minimum-norm attack point that changes the label, which most of the methods you consider do not do. You do include the full curve in Figure 4 (please make the font bigger though). Under the full curve it seems the improvement over other methods is minimal, and could possibly be due to hyperparameter tuning.

Finally, more minor but the Gaussian derivation does not make sense. It is unlikely that your norms actually follow a Gaussian distribution. A more appropriate claim would be that you are constraining or penalizing the expected norm of the perturbations. I would remove the math part as it does not add anything to the paper (and is also wrong as per preceding comment) and focus more on how you actually construct the generator network (this is only briefly discussed in the appendix and not at all in the main text, even though it is a key point to getting the method to work). You could also use the space for more detailed experiments, following best practices as in https://arxiv.org/abs/1902.06705 to ensure that your evaluation is sensible.

**Experience Assessment:**

I have published in this field for several years.

**Review Assessment: Checking Correctness Of Derivations And Theory:**

I assessed the sensibility of the derivations and theory.

**Review Assessment: Checking Correctness Of Experiments:**

I assessed the sensibility of the experiments.

**Review Assessment: Thoroughness In Paper Reading:**

I read the paper at least twice and used my best judgement in assessing the paper.

---

> ### Author Response · Authors · 2019-11-11
> **Response to Reviewer #3 (2 of 2)**
>
> 2.	Evaluation Method
>
> We had contemplated deeply on how we could fairly compare our algorithm with FGM and PGD adversarial training methods. You could find a detailed explanation on our evaluation method in Section 4.2. To fairly evaluate adversarial training algorithms, we have to check two things: 1) the robustness metric that needs to be used, 2) the hyperparameter for each adversarial training algorithm.
>
> First, we set the robustness metric as the average distance of the successful adversarial perturbations of the Carlini & Wagner L2 attack. We believe that this is a very fair metric because C&W attack is considered as the most powerful adversarial attacks so far, and this attack has not been used in the training process of all the adversarial training algorithms used in our experiments. This method also coincides with the recommended method in the paper mentioned in your comment. (https://arxiv.org/abs/1902.06705, page5 & page12).
>
> Secondly, we admit that it is challenging to simply match the hyperparameters for each algorithm, because FGM and PGD constrain the L2 norm while ours penalizes the L2 norm as you have mentioned in your comment. To solve this issue, we matched the benign accuracy (accuracy with clean examples) for each network. Only when the benign accuracy for each network is matched prior to the evaluation of the robustness metric, a fair comparison is made to decide which adversarial training algorithm performs the best.
>
> In Table 1, we first matched the benign accuracy for each network to the benign accuracy of the baseline network, and then compared the robustness of each network against various attacks. FGM and MIM are norm-constrained attacks, while Deepfool and C&W attack is unbounded norm attacks. Therefore, a robust network has higher accuracy in FGM and MIM, and higher L2 Mean of the distortion in Deepfool and C&W attacks. We will revise the table in the revision, since the current one may be confusing.
>
> In Figure 4, the three algorithms were compared using various benign accuracy rates (68%, 66%, and 63%, respectively). For all considered cases, the network trained with our algorithm was shown to be more robust than the networks trained with the other two algorithms.
>
> In comparing different algorithms, the evaluation method depicted in Table 1 and Figure 4 has a limitation in that the networks can only be compared under certain hyperparameter.  It was desirable to visualize the comparison of the three algorithms under various hyperparameters. In Figure 5, the plot on the left shows the benign accuracy vs. robustness (with C&W attack) curve by adjusting hyperparameters for each adversarial training algorithms. Under the same benign accuracy conditions, our algorithm shows higher robustness than the other two algorithms. It is believed that plots in Figure 5 is a fairer comparison method than Table 1 or Figure 4, and it shows the superior performance of our algorithm.
>
>
> 3.	Gaussian derivation
>
> We did not assume that the perturbation norm follows the Gaussian distribution, but we just assumed that the class probability of a classifier with perturbation norm follows a normal distribution function. In fact, humans also find it difficult to distinguish objects properly in foggy or cloudy weather, and we have depicted this phenomenon mathematically. In other words, the conventional adversarial loss assumes that the class probability should be one for “small” perturbation norm and zero for the others. On the other hand, our adversarial loss assumes that the class probability is one for “zero” perturbation norm, and it gradually decreases with the perturbation norm.
>
> We appreciate your kind review again, and we are currently revising the manuscript based on what you pointed out.
>
> [1] Tramèr, Florian, Alexey Kurakin, Nicolas Papernot, Ian Goodfellow, Dan Boneh, and Patrick McDaniel. "Ensemble Adversarial Training: Attacks and Defenses." (2018).
> [2] Anish Athalye, Nicholas Carlini, and David Wagner. “Obfuscated gradients give a false sense of security: Circumventing defenses to adversarial examples” (2018)

---

> ### Author Response · Authors · 2019-11-11
> **Response to Reviewer #3 (1 of 2)**
>
> I would like to thank you for your valuable review. We agree with your comments mostly, and here are our initial responses to your comments.
>
>
> 1.	The reason why neural net performs better than PGD or FGM
>
> As you have claimed in your comments, the fact that there is a gradient masking problem in conventional adversarial training is a strong evidence that a classifier can easily be overfitted by the attacks from FGM or PGD. First, we admit that PGD and FGM attacks are somewhat adaptive as they depend on the current model parameters. But what we would like to emphasize here is that although these algorithms use the gradients of model parameters, the attack ‘method’ that uses the gradients is fixed. For example, in FGM, adversarial perturbation is generated by normalizing the gradient of the data sample, and then multiplying by the $\epsilon$. A classifier can be easily made robust against adversarial examples generated by these fixed methods. The following are two possible methods.
>
> CASE 1. The classifier is trained to gradually reduce the gradient at the data points. Gradually reducing the gradient prevents FGM from generating meaningful adversarial example from the gradient (gradient masking issues).
>
> CASE 2. The classifier can only be trained to reduce the loss under the surface of the $\epsilon$-norm ball centered around the data point. So, the network would be robust against adversarial examples generated by FGM, but could still be very vulnerable against adversarial examples located inside the $\epsilon$-norm ball.
>
> Through the above methods, a classifier can be easily made robust (overfitted) against FGM and PGD attacks, but it becomes more vulnerable to other powerful attacks such as Carlini & Wagner attacks (Please refer to the red curve on the right plot in Figure 3). Through our experiments under FGM adversarial training, we observed the CASE 1 overfitting problem when $\epsilon$  is small, and the CASE 2 overfitting problem when $\epsilon$  is large. The gradient masking problem such as CASE 1 has also been discussed by other papers on adversarial training ([1], [2]). On the other hand, our algorithm continuously updates the parameters of a generator to maximize the classifier loss, which produces more general (adaptive) attacks (not restricted to a particular attack method) to the classifier. This means that a classifier cannot easily get overfitted against attacks from a generator. In the revision, we will include all the justifications explained above and additional experiment results to support this.

---

> > ### Comment · AnonReviewer3 · 2019-11-12
> > **Need data to justify the argument**
> >
> > Thank you for your response regarding the neural net approach vs. PGD. I do not find it convincing because it only provides a verbal argument for why a method *might* work better than PGD. A more convincing argument would include actual empirical data to support the claims.
> >
> > In addition, it's not clear that Case 1 arbitrates in favor of the neural net approach, since the neural network approach could also have difficulties given uninformative gradients. And Case 2 only affects FGM, not PGD.
> >
> > As a starting point for constructing an empirical case, see my earlier comment:
> >
> > "If we believe this story, then actually the neural net attacker should already work well just as an attack (i.e. just train it to generate good attacks against a fixed model for some set of examples). I would find this quite interesting if true but unfortunately it wasn't explored in the paper. One way to do it would be to take models that are believed to be robust based on attacking them with PGD, and then show that the neural net attacker is substantially more accurate."

---

> > > ### Author Response · Authors · 2019-11-13
> > > **Thank you for your helpful suggestion**
> > >
> > > Thank you for your helpful suggestion. Based on your feedback, we conducted the following additional experiments with CIFAR-100. To verify the attack performance of the neural network, we used two fixed classifier networks adversarially trained with FGM ($\epsilon$ = 0.5) and PGD ($\epsilon$=0.5), respectively. For each classifier network, we compared the performance of FGM attack, PGD attack, and PGN (proposed) attack. PGN was fully trained on the training dataset. We plotted the accuracy of the classifiers for these three attacks in terms of average perturbation distortion. For each model, it is observed that our algorithm is capable of much more powerful attack than PGD and FGM. This difference in performance clearly justifies the outstanding defense performance of the adversarial training with PGN in Section 4.3. These experimental results are temporarily available on our github page https://github.com/ATPGN/ATPGN and will soon be added to our revision paper.
> > >
> > > We believe that the reason for the improved performance is the flexibility of the generator’s attack method. Our PGN generator can be trained to fully exploit incoming gradients, images, and labels to generate adversarial perturbation optimized for the classifier. As for FGM or PGD, they have a fixed norm, with a single attack direction towards gradient. Now with PGN, it has no limitation on both direction and size, and can be transformed into an optimized perturbation for the classifier.
> > >
> > >  We will add these experimental results of the generator's attack performance to Section 4 in the revision to further strengthen our justification. Thank you again for your thoughtful comments. We will notify you when the revision paper is uploaded.

---

### Author Response · Authors · 2019-11-15
**Revised draft is now available**

Thank you very much for all your constructive comments and suggestions. We have uploaded our revised paper. We tried to address as many your feedback and opinions as possible. The following changes were made in the revision:

1) We have moved the ‘THEORETICAL BACKGROUND’ in Section 3 to Appendix, and used the remaining space for more detailed argument why our algorithm may perform better than the conventional methods. The related content has been added to the last paragraph of Section 3.2.

2) To better prove our argument in 1), we have conducted additional experiments on which attack method most effectively attacks a given classifier network that is adversarially trained with FGM or PGD. The experimental results showed that our generator network is capable of more effective attack compared to FGM or PGD. The results of this experiment are added in Section 4.3.

3) We have visualized the adversarial examples  for various perturbation norms generated from the proposed generator network in Appendix C. Also in Appendix D, we conducted an ablation test on which input among the three (gradient, image, and label) helps the most in our experimental results.

4) Table 2 and Table 3 are slightly changed to make them more readable. FGM and MIM are bounded-norm attack methods, meaning that the higher the accuracy, the more robust the network. Deepfool and Carlini & Wagner $L_2$ attack are unbounded-norm attacks, which indicates that the higher the average norm of the perturbation is, the more robust the network gets. For better readability, we masked all the data that is not related to the actual robustness metric.

5) We have increased the size of the figures and captions that were not legible.

We would like to thank you again for your valuable feedback. We were able to complete a more persuasive paper thanks to all your helpful comments.

---

### Decision · Program_Chairs · 2019-12-19

**Decision:**

Reject

**Comment:**

This paper proposes to use the GAN (i.e., minimax) framework for adversarial training, where another neural network was introduced to generate the most effective adversarial perturbation by finding the weakness of the classifier. The rebuttal was not fully convincing on why the proposed method should be superior to existing attacks.